# Co-Targeting of DTYMK and PARP1 as a Potential Therapeutic Approach in Uveal Melanoma

**DOI:** 10.3390/cells13161348

**Published:** 2024-08-14

**Authors:** Sylwia Oziębło, Jakub Mizera, Agata Górska, Mateusz Krzyziński, Paweł Karpiński, Anna Markiewicz, Maria Małgorzata Sąsiadek, Bożena Romanowska-Dixon, Przemysław Biecek, Mai P. Hoang, Antonina J. Mazur, Piotr Donizy

**Affiliations:** 1Department of Cell Pathology, Faculty of Biotechnology, University of Wroclaw, 50-383 Wroclaw, Poland; 2Department of Clinical and Experimental Pathology, Wroclaw Medical University, 50-556 Wroclaw, Poland; 3Faculty of Mathematics and Information Science, Warsaw University of Technology, 00-662 Warsaw, Polandprzemyslaw.biecek@gmail.com (P.B.); 4Department of Genetics, Wroclaw Medical University, 50-368 Wroclaw, Polandmaria.sasiadek@umw.edu.pl (M.M.S.); 5Department of Ophthalmology and Ocular Oncology, Faculty of Medicine, Jagiellonian University Medical College, 31-008 Krakow, Polandromanowskadixonbozena1@gmail.com (B.R.-D.); 6Department of Pathology, Massachusetts General Hospital, Harvard Medical School, Boston, MA 02114, USA

**Keywords:** DTYMK, immunohistochemistry, inhibitors, PARP1, uveal melanoma, BAP1, intraocular tumor

## Abstract

Uveal melanoma (UM) is the most common primary intraocular tumor in adults, with no standardized treatment for advanced disease. Based on preliminary bioinformatical analyses DTYMK and PARP1 were selected as potential therapeutic targets. High levels of both proteins were detected in uveal melanoma cells and correlated with increased tumor growth and poor prognosis. In vitro tests on MP41 (BAP1 positive) and MP46 (BAP1 negative) cancer cell lines using inhibitors pamiparib (PARP1) and Ymu1 (DTYMK) demonstrated significant cytotoxic effects. Combined treatment had synergistic effects in MP41 and additive in MP46 cell lines, reducing cell proliferation and inhibiting the mTOR signaling pathway. Furthermore, the applied inhibitors in combination decreased cell motility and migration speed, especially for BAP1-negative cell lines. Our hypothesis of the double hit into tumoral DNA metabolism as a possible therapeutic option in uveal melanoma was confirmed since combined targeting of DTYMK and PARP1 affected all tested cytophysiological parameters with the highest efficiency. Our in vitro findings provide insights into novel therapeutic avenues for managing uveal melanoma, warranting further exploration in preclinical and clinical settings.

## 1. Introduction

Uveal melanoma (UM), a rare malignancy arising from melanocytes of the uveal tract of the eye predominantly within the choroid of the eye, represents a significant clinical challenge [1]. Besides the recent implication of a few novel molecularly targeted therapies, the presence of distant metastases is associated with a poor prognosis (18% 5-year survival in metastatic cases vs. 84% for ocular-localized tumors) [2]. In addition, there is no uniform treatment for advanced disease, and immunotherapy has not shown a durable response in contrast to the front-line treatment of advanced cutaneous melanomas [3]. Hence, there is an urgent need to search for other targets to create novel molecularly tailored therapies.

Based on our preliminary in silico analyses in the present study, which included an iterative selection of independent markers associated with survival and toxicity to cancer cell lines, we selected two potential protein targets. These proteins are DTYMK (deoxythymidylate kinase) and PARP1 (poly (ADP-ribose) polymerase 1), and both are responsible for deoxyribonucleic acid (DNA) metabolism. Moreover, their potential inhibition may significantly reduce or diminish tumor growth (hypothesis of the double hit into tumoral DNA metabolism).

DTYMK is the kinase playing a key role in DNA synthesis via dTMP phosphorylation catalysis, and its overexpression is related to unfavorable prognoses in several types of human cancers (e.g., lung and liver cancer) [4,5]. DNA biosynthesis requires a balanced supply of deoxyribonucleotide triphosphates (dNTPs), including dTTP. The activity of DTYMK contributes to the regulation of dTTP levels by producing dTDP, which can be further converted to dTTP through additional enzymatic reactions. Proper regulation of dNTP pools is essential for DNA replication fidelity. Hence, it is considered a crucial protein for maintaining genome stability and impacting immune system function [6]. Higher DTYMK levels can compromise NK cell activity suppressing anti-cancer immune response [4]. DTYMK has also been identified as a competitive binder of miR-378a-3p, which leads to the preservation of MAPKAPK2 activity. Consequently, this interaction triggers activation of the phospho-HSP27/NF-kB signaling axis, which plays a pivotal role in inducing drug resistance, enhancing cell proliferation, and promoting tumor infiltration with macrophages [7]. Furthermore, in some malignancies, DTYMK was found to modulate cell migration [8]. Our bioinformatical analysis conducted in the present study and based on the Cancer Genome Atlas (TCGA) datasets has shown high DTYMK gene expression significantly correlated with shorter overall survival of UM patients, independently from chromosome 3 status (which is a crucial unfavorable prognosticator), making this enzyme a promising target for novel molecularly tailored therapies (a specific DTYMK inhibitor is commercially available).

PARP1 is the pleiotropic enzyme that serves many functions, such as chromatin modification and transcriptional regulation, but among the most significant ones is undeniably its role in initiating the DNA repair process [9,10]. Due to its pivotal role in maintaining genome stability, its increased level is an important poor prognostic factor in a wide range of cancers (including skin and mucosal melanomas) [11]. The DNA repair process induced by PARP1 takes place in at least four mechanisms, including homological recombination (HR), base excision repair (BER), non-homologous end joining (NHEJ), and processing of the Okazaki fragments. Cancer cells with impaired HR pathways exhibit a deficiency in double-strand breaks (DSBs) repairing. These cells exploit alternative repair pathways such as BER and NHEJ to sustain their viability. As a compensatory mechanism, the activity of PARP1, responsible for triggering the BER and NHEJ pathways, is frequently increased in HR-deficient cancer cells. Consequently, these cells exhibit heightened sensitivity to PARP1 inhibition due to the impairment in HR in combination with suppressed BER/NHEJ pathways. This inhibition leads to an accumulation of DSB lesions, which is particularly detrimental to actively replicating tumor cells, as unrepaired breaks can result in the lack of essential genes in daughter cells. In contrast, wild-type (HR-proficient) cells are unaffected by PARP1 inhibitors since they can still effectively repair DNA damage through the high-fidelity HR pathway [10]. PARP1 was also found to be involved in inflammatory processes by regulating cytokines, adhesion factors, transcriptional factors, and other inflammatory mediators such as TNFα and NOS [12]. Our preliminary clinical studies on UM presented in this article have shown high PARP1 immunoreactivity significantly correlates with high proliferation activity of tumoral cells and shorter overall and metastasis-free survival of UM patients. These findings indicate the potential important impact of PARP1 inhibition on diminishing UM progression.

The aim of this study was the analysis of DTYMK and PARP1 expression profiles using in vitro models based on two UM cell lines [BAP1-positive (MP41) and BAP1-negative (MP46)] and the determination of DTYMK and PARP1’s co-inhibition impact on parameters related to invasiveness and survival of UM cells. To date, there are no reports regarding DTYMK and PARP1 inhibition on UM cell lines and their role in the progression of this rare subtype of human melanoma.

## 2. Materials and Methods

### 2.1. Clinical and Histologic Information

This study included 164 patients diagnosed with UM at the Department of Ophthalmology and Ocular Oncology, Medical College, Jagiellonian University in Krakow, Poland. The ages of the patients ranged from 18 to 86 years (median, 60 years). The range of follow-up was 1 to 195 months (median, 76 months). A total of 49/164 (30%) developed metastasis. A total of 101 of 164 (62%) patients died, and 48 of these 101 (48%) deceased patients had documented metastatic uveal melanoma. Six patients (6%) died from other causes. The clinical information was extracted from the medical records, and histopathologic parameters were assessed by examining the tissue sections of enucleation specimens. This information has been presented in detail in previous publication [13].

### 2.2. RNA Sequencing Analyses

Preprocessed TCGA RNA-seq data (raw counts) of 75 UVM bulk tumor samples and clinical data were downloaded from NCI Genomic Data Commons (GDC) [14,15]. For TCGA UVM data, lowly expressed genes were discarded and then were transformed by log(tpm) (transcript per million) values using in-house R script.

Preprocessed single-cell RNA-seq profiling (raw counts) of 11 patient-derived uveal melanoma samples (103,603 cells in total), including 8 primary and 3 metastatic samples, was downloaded from Gene Expression Omnibus (GEO) database, accession number: GSE139829 [16].

Single-cell RNA-seq data was transformed to Seurat object followed by filtering for cells containing a minimum of 200 and no more than 8000 unique genes and containing less than 10% mitochondrial genome. Filtered data were normalized using global-scaling normalization method “LogNormalize” (scale.factor = 10,000) in Seurat Bioconductor package (version 4.4.0) [17]. To visualize single-cell RNA-seq data, we selected the top 2000 variable features identified using the vst method. Next, data were scaled, and principal component analysis (PCA) was performed. Thereafter, the elbow plot method was used to select number of principal components for Unifold Manifold Approximation and Projection (UMAP) non-linear dimensionality reduction followed by calculating the k-nearest neighbor graph and Louvain clustering [17]. FeaturePlot() and DotPlot() functions were used for visualizing gene(s) expression in low-dimensional space.

### 2.3. Immunohistochemistry

Tissue microarrays comprised of 2 mm tissue cores from each paraffin-embedded tumor were constructed for DTYMK and PARP1 immunohistochemistry (IHC) studies. Representative tissue material was available for 156 patients. Immunohistochemistry was performed on five-micrometer-thick formalin-fixed, paraffin-embedded tissue sections using standard peroxidase IHC techniques, heat-induced epitope retrieval buffer, and primary antibodies against DTYMK (EPR10163, 1:100, Abcam, Cambridge, United Kingdom) and PARP1 (sc-74470 (B10), dilution: 1:50, Santa Cruz Biotechnology, Dallas, TX, USA). EnVision FLEX/HRP (DAKO, Santa Clara, CA, USA) and Liquid Permanent Red (DAKO) were used as detection systems for DTYMK and PARP1, respectively.

Scoring of DTYMK and PARP1 immunostains was performed using the H-score. The score is obtained with the formula: percentage of tumoral cells with weak reactivity × 1 + percentage of tumoral cells with intermediate reactivity × 2 + percentage of tumoral cells with high reactivity × 3, giving a range of 0 to 300. DTYMK and PARP1 H-scores were dichotomized into two groups using the method proposed by Contal and O’Quigley, which is based on the log-rank test statistic. Overall survival was used independently for both H-scores. The optimal cutpoint for DTYMK H-score was 240 and 25 for PARP1. Detailed immunohistochemical parameters for BAP1 and Ki67 have been described in detail in previous publication [18].

### 2.4. Statistical Analyses

Statistical analyses on external RNA sequencing and single-cell RNA sequencing data were performed in R 4.3.1 and Bioconductor 2.60. All reported *p*-values were corrected for multiple testing using the Benjamini and Hochberg method.

To select robust gene markers associated with overall survival (OS) of UM patients independently from potentially interacting covariates (age, gender, chromosome 3 status (disomic, monosomic), and stage), we utilized cross-validation, forward selection, and partial likelihood of the Cox model implemented in rbsurv R package version 2.62.0 [19]. To examine the potential importance of selected markers for viability of cancer cell lines, we assessed the Cancer Dependency Map portal [20].

Determination of optimal cutpoint for *DTYMK* or *PARP1* genes log(tpm) expression values with respect to which cohort patients were stratified into two groups (“high” expression and “low” expression) was performed by using the maximally selected rank statistics [21]. To assess association of DTYMK or PARP1 expression with overall survival (OS), we used multivariate analysis using the Cox regression to adjust for potentially interacting covariates (age, gender, chromosome 3 status (disomic, monosomic), and stage) in “survival” and “survminer” R packages versions 3.7-0 and 0.4.9 respectively [22]. Validity of the Cox model assumptions was assessed with cox.zph() function. The psych package version 2.4.6.26 was used to calculate Pearson’s correlation between DTYMK, PARP1, and BAP1. Significant correlation was defined as absolute correlation coefficient ≥ 0.3 and adjusted *p*-value ≤ 0.05.

Statistical analyses of our cohort of patients were conducted within the R environment [23] utilizing the survival [24] and survminer [25] packages versions 4.4.1, 3.6.4 and 0.4.9 respectively. Overall survival (OS) served as the primary endpoint, defined as the number of days from initial diagnosis to death by any cause. Correlations between numerical covariates were assessed using Pearson’s correlation coefficient. For group comparisons of covariate values, the Wilcoxon–Mann–Whitney test was used. Dichotomization of continuous covariates was performed using the method proposed by Contal and O’Quigley [26]. The log-rank test was utilized to test statistical significance of differences in survival outcomes between patient groups, applicable to both two-group and multi-group comparisons. Kaplan–Meier curves were then generated to visually depict these survival outcomes. In all analyses, a *p*-value threshold of less than 0.05 was set for statistical significance, given the absence of multiple testing.

In in vitro analyses, all data are given as means ± standard deviations (SD), and their significance was determined with one-way ANOVA followed by Dunnet’s post hoc test. The significance threshold was set at *p* ≤ 0.05 (*), *p* ≤ 0.01 (**), *p* ≤ 0.001 (***), *p* ≤ 0.0001 (****). All statistical analyses and preparation of bar charts or plotting of graphs were conducted using GraphPad Prism 9 software (GraphPad Version 9.9.1 (GraphPad Software Inc., San Diego, CA, USA) (Dotmatics version 6.1).

### 2.5. Cell Culture and Reagents 

The human uveal melanoma MP41 (BAP1 positive) and MP46 (BAP1 negative) cell lines were purchased from the American Type Culture Collection (ATCC^®^ Manassas, VA, USA). Cells were cultivated in RPMI 1640 medium (Gibco^TM^ Waltham, MA, USA) supplemented with 20% (*v*/*v*) fetal bovine serum (FBS), 1% (*v*/*v*) glutamine, and antibiotics (10,000 U/mL penicillin, 10,000 µg/mL streptomycin, 25 µg/mL Amphotericin B) (Gibco™, accordingly to ATCC^®^ recommendation for culturing those cell lines). Cell culture was maintained in tissue culture flasks recommended explicitly by the ATCC^®^ (431464U, Corning^®^ Corning, NY, USA) under standard conditions at a temperature of 37 °C, gas mixture of 5% CO_2_, and 95% humidified air, sub-cultured biweekly. The sub-culture process used 0.25% trypsin/0.05% EDTA solution of pH 7.2 (IITD PAN, Wrocław, Poland).

### 2.6. Treatment of Cells with Inhibitors

PARP1 inhibitors, specifically AG-14361, Niraparib tosylate (MK-4827), Pamiparib (BGB-290), and Rucaparib camsylate were purchased from Selleckchem, while DTYMK inhibitor Ymu1 was obtained from Sigma-Aldrich, Saint Louis, MO, USA. Experimental setup for the PARP and DTYMK inhibition involved the treatment of the cells either with inhibitors individually (either the DTYMK inhibitor or one of PARP1 inhibitors at a time) as well as combined conditions of DTYMK inhibitor with Pamiparib/PARP1 inhibitor, respectively.

The concentrations of the inhibitors employed in all assays were determined based on preliminary viability experiments (described in the Section 3) and were customized to align with the sensitivity profile of the respective cell lines. For XTT (viability), migration, assays, and Western blot analysis, cells were exposed to Pamiparib at concentrations of either 10 µM or 25 µM, Ymu1 at concentrations of either 10 µM or 15 µM, or various combinations of inhibitors (10 µM Pamiparib with 15 µM Ymu1, 25 µM Pamiparib with 10 µM Ymu1, or 25 µM Pamiparib with 15 µM Ymu1). Additionally, a control group comprised cells incubated solely with the addition of respective amounts of DMSO (Sigma-Aldrich, Saint Louis, MO, USA) (vehicle control), which served as the solvent for the inhibitors.

### 2.7. Viability Evaluation and Synergy Assessment

The CyQUANTTM XTT Cell Viability Assay (ThermoFisher Scientific, Waltham, MA, USA), a colorimetric assay employed to assess cell count predicated on their metabolic activity, was implemented according to the manufacturer’s prescribed protocol. The assay was executed in 96-well plates (the cells were seeded in a density of 7500 cells per well). Following 72 h period of cell treatment with an inhibitor, the XTT labeling mixture was added to the test samples. After a 4 h incubation period with XTT reagent, absorbance readings were taken, and the acquired data were subsequently subjected to background correction. The average cellular viability was expressed as a percentage decrease in viability (absorbance) compared to vehicle control cells (designated as 100% viability). Each experimental condition was conducted in quadruplicate for each specific cell line. Synergy assessment was conducted by processing cytotoxicity data in SynergyFinder tool available online (developed by the Network Pharmacology for Precision Medicine in the Research Program of System Oncology, Faculty of Medicine at University of Helsinki, Helsinki, Finland) [27].

### 2.8. Cell Proliferation Analysis

The Click-iTTM EdU Proliferation Assay for Microplates (ThermoFisher Scientific, Waltham, MA, USA), which relies on incorporating the nucleoside analog EdU into the newly synthesized DNA, was employed following the manufacturer’s protocol. The assay was performed in 96-well plates containing 7500 cells per well. After 48 h of cells’ treatment with inhibitors, the 10 µM EdU reagent was administered. After another 24 h of incubation, EdU was detected following the manufacturer’s instructions, and fluorescence readings were recorded using a microplate reader (GloMax^®^ Discover—Promega, Madison, WI, USA) at 580/640 nm wavelengths. All experimental conditions were executed in quadruplicate for each cell line.

### 2.9. Western Blot Analysis

A total of 24 h after cell seeding, the cells were administered with designated concentrations of inhibitors and treated further for 24 h or 48 h in tested conditions. To obtain protein samples, cells were harvested and lysed by the addition of lysis buffer (50 mM Tris-HCl pH 7.4, 5% *v*/*w* SDS, 8.6% *w*/*v* sucrose, 1 mM DTT, 74 mM urea), supplemented with protease and phosphatase inhibitors cocktails (Sigma-Aldrich) diluted 1:100 prior to use. The protein concentration of the lysates was determined using Pierce™ BCA Protein Assay Kit (ThermoFisher Scientific) following the manufacturer’s protocol. An equal protein amount (15 µg) from each sample, along with the protein mass marker (PageRuler™ Prestained Protein Ladder 10–170 kDa, ThermoFisher Scientific), was subjected to separation using 12.5% polyacrylamide gel electrophoresis in the presence of sodium dodecyl sulfate (SDS-PAGE) following the method described by [28]. The separated proteins were subsequently transferred to nitrocellulose sheets, as described elsewhere [29]. The half-quantitative assessment of protein loading among the wells was assessed with 0.1% (*w*/*v*) Ponceau S (in 5% (*v*/*v*) acetic acid) staining of the membrane (Sigma-Aldrich). The Ponceau S staining was washed out of the membranes with TBS-T (Tris-buffered saline with 0.1% (*v*/*v*) Tween 20), and then membranes were blocked in 5% (*w*/*v*) non-fat milk in TBS-T. In the immunoblotting procedure, the membranes were incubated overnight at 4 °C with respective primary antibodies (as listed in Appendix A) diluted in blocking solution. Secondary antibodies conjugated with horseradish peroxidase, goat anti-rabbit, and goat anti-mouse (as listed in Appendix A) were utilized according to the manufacturer’s protocols (1 h of incubation under gentle agitation, diluted as stated in Appendix A in non-fat milk in TBS-T). Immunoblots were developed using the Clarity Western ECL Substrate (Bio-Rad, Hercules, CA, USA), scanned with ChemiDoc (BioRad), and analyzed with ImageLab software (ver. 6.0, Bio-Rad). At least three independent experiments were conducted.

### 2.10. Single-Cell Migration Assay 

Cells were seeded in the 96-well plate (ImageLock from Sartorius, Gottingen, Niedersachsen, Germany) (at a density of 4500 cells per well). A total of 24 h after cell seeding, the culture medium was replaced with the fresh one, containing the previously indicated concentrations of inhibitors. The plates were then placed in the IncuCyte^®^ system (Sartorius, Goettingen, Germany) with IncuCyte Zoom Software (Sartorius, Goettingen, Germany version 2018A, version details: 20181.1.6628.28170) ) for the 72 h incubation, during which the system captured pictures of the cells at 2-hour intervals. These datasets were subsequently utilized to gather measurements of the distance the single cell covers, its velocity, and single-cell trajectories. Measurements were taken for ten cells per biological repetition using the manual tracking plug-in for ImageJ version 1.53 (Manual Tracking plugin (ImageJ, F. Cordelieres, Institute Curie, Paris, France). Directionality was calculated as described elsewhere [30,31].

### 2.11. Immunofluorescence Techniques and Microscopy

Details on reagents used for stainings and respective dilution factors for the following experiments were summarized in Appendix A. Cells were seeded on sterile coverslips and cultured in standard conditions. After 48 h post-seeding, Mitotracker^®^ (ThermoFisher Scientific) was introduced to a final concentration of 100 nM. Following 30 min incubation, cells were fixed with 4% (*w*/*v*) formaldehyde (Sigma-Aldrich) in PBS for 20 min at room temperature and permeabilized for 6 min with 0.1% Triton X-100 (Sigma-Aldrich) in PBS. Next, coverslips were blocked for 30 min with 1% (*w*/*v*) bovine serum albumin in PBS. Incubation with primary antibodies targeting PARP1 and DTYMK was set overnight at 4 °C. Following day, coverslips were probed with secondary antibodies: anti-mouse conjugated with Alexa Fluor™ 647 or Alexa Fluor™ 568 antibodies (respectively) or anti-rabbit antibodies conjugated with Alexa Fluor™ 488 (ThermoFisher Scientific). The cells were additionally stained with phalloidin conjugated with Alexa Fluor™ 568 (ThermoFisher Scientific) and Hoechst 33,342 (Thermo Fisher) to detect filamentous actin (F-actin) and DNA, respectively. Incubation was carried out in a dark environment for one hour at room temperature. Finally, coverslips were mounted with Dako fluorescent mounting medium (Dako, Santa Clara, CA, USA) on microscope slides and imaged with Leica TCS SP8 using LAS X software (Leica Application Suite X (LasX) version 3.7.4.23463 2020 (Leica Microsystems CMS GmbH, Wetzlar, Germany)).

## 3. Results

### 3.1. Selection of DTYMK and PARP1 as Potential Therapeutic Targets Using In Silico Analyses

Using multiple cross-validations on UM TCGA data, we identified *DTYMK* as a gene independently associated with poor survival (Figure 1A). We also tested the prognostic significance of PARP1 as a possible promising co-targeting partner of DTYMK in the inhibition of DNA metabolism. Survival analysis of the TCGA cohort revealed a significant negative impact of PARP1 expression on long-term prognosis (Figure 1B). Subsequently, we assessed the expression of DTYMK and PARP1 and some other UM-related markers in the single-cell RNA-seq UM dataset. We found intermediate expression levels of DTYMK and PARP1 in tumor cells (Figure 1C). Moreover, DTYMK and PARP1 were also variably expressed in UM tumor cells between patients (Figure 1D,E). Using the data from ~50,000 UM malignant cells from primary tumors, we found a very weak, insignificant correlation between DTYMK and BAP1 (correlation coefficient = −0.003), PARP1 and DTYMK (correlation coefficient = 0.12), and PARP1 and BAP (correlation coefficient = −0.02).

### 3.2. DTYMK and PARP1 Expression in Uveal Melanoma Patients

Cytoplasmic DTYMK scores range from 0 to 300 (median: 240, mean: 220, SD: 82), and nuclear PARP1 H-scores range from 0 to 210 (median: 5, mean: 30, SD: 48). DTYMK and PARP1 H-scores were dichotomized into two groups using the method proposed by Contal and O’Quigley, which is based on the log-rank test statistic. The optimal cut point for the DTYMK H-score was 240 and 25 for PARP1. Overexpression of DTYMK and PARP1 was observed in 51.3% (80/156) and 31.4% (49/156) of patients (Figure 2A–D). Overexpression of both DTYMK and PARP1 was observed in 21.2% (33/156). Statistical analysis revealed a very weak correlation between DTYMK and PARP1 expression (Figure 2E). Moreover, there were no significant differences between DTYMK expression values for BAP1-retained and BAP1-loss tumors (Figure 2F). Taken together, these results indicate the independent role of DTYMK and PARP1 in the pathobiology of uveal melanoma, regardless of BAP1 status.

### 3.3. Overexpression of DTYMK and PARP1 Protein Levels Correlated with Worst Long-Term Prognosis

High levels of DTYMK and PARP1 were significantly correlated with the high proliferation potential of primary tumors measured by enhanced mitotic index and elevated Ki-67 immunoreactivity (*p* = 0.01, *p* < 0.001 and *p* < 0.001, *p* = 0.002, respectively, for DTYMK and PARP1). Kaplan–Meier analysis showed a negative impact of enhanced expression of DTYMK as a single marker (Figure 2G), as well as in combination with PARP1 (Figure 2H). Patients with overexpression of both proteins were characterized by the worst long-term prognosis.

### 3.4. Validation of the Presence of DTYMK and PARP1 in Studied Cell Lines

Uveal melanoma (UM) cells are characterized by the presence/absence of BAP1 protein and mutation in the GNAQ/GNA11 genes [32]. For this study, it was decided to utilize two cell lines representing two major genetic and phenotypic background variants to address these two dominant features of UM cells. The MP41 cell line is characterized by the production of BAP1 protein and possession of c.626A>T (p.Gln209Leu) mutation in the GNA11 gene. On the contrary, the MP46 cells do not produce BAP1 protein and have mutated the GNAQ gene at c.626A>T (p.Gln209Leu) [32].

Throughout Western blot analysis, the presence of PARP1 and DTYMK was confirmed both in MP41 and MP46 cell lines (Figure 3A). The signal for the DTYMK protein was observed at approximately 24 kDa, which corresponds to the first isoform of the protein [33]. No other DTYMK isoforms were found in the tested cells. For PARP1, a single band of 113 kDa was observed. Given the absence of non-target bands, it is safe to assume significant specificity of the antibodies.

### 3.5. Determination of the Cellular Localization of DTYMK and PARP1 

In the next step, we evaluated the morphology of the studied cells and the subcellular localization of DTYMK and PARP1 enzymes. F-actin staining highlighted evident morphological differences between the tested uveal melanoma cell lines: MP41 cells are elongated and spindle-shaped, while MP46 cells appear slightly elongated and star-shaped. This is in agreement with the previous report [32]. Going further, the subcellular distribution of DTYMK and PARP1 in uveal melanoma cells was determined through immunocytochemical analyses. We confirmed canonical localization of the tested proteins: nuclear for PARP1 and cytoplasmic for DTYMK, in both MP41 and MP46 cell lines (Figure 3B,C). Notably, the co-staining of mitochondria provides even more proof of the presence of the mitochondrial fraction of the DTYMK along the cytoplasmic one [34]. Interestingly, the tendency for mitochondrial localization differed between the cell lines, being more prominent in MP46 cells (Figure 3C).

### 3.6. The Cytotoxic Effect of DTYMK and PARP1 Inhibitors on the Uveal Melanoma Cells

Since we observed the overexpression of DTYMK and PARP1 in tumors derived from patients with uveal melanoma, we decided to test the influence of inhibitors of these enzymes in the in vitro model. To verify the impact of individual PARP1 and DTYMK inhibitors on MP41 and MP46 cell viability, an XTT assay was performed. For the cytotoxicity assay, we analyzed the cells 72 h after administration of drugs because the doubling time of MP41 and MP46 is 41 h and 110 h, respectively [32].

The final percentage of DMSO in vehicle controls each time was 2%, and it was the same amount as in the highest doses of inhibitors. The initial screening included four PARP1 inhibitors, i.e., tosylate AG-14361, niraparib (MK-4827), pamiparib (BGB-290), and rucaparib camsylate in a wide range of concentrations (0.1–50 µM). Among the tested PARP1 inhibitors, only pamiparib significantly affected the cell viability of MP41 cells and thus has been chosen for further experiments (Appendix A). None of the tested PARP1 inhibitors showed a cytotoxic effect on MP46 cells. Next, we determined the IC50 value for pamiparib on both cell lines. In the case of MP41 cells, the IC50 value for pamiparib was 29.07 µM (Figure 4A,B). Determination of the IC50 value for the MP46 cell line was not possible due to the low cytotoxic effect of this inhibitor on MP46 cells (Figure 4C). As for the DTYMK inhibitor (Ymu1), the 2–30 µM concentration range was tested (Figure 4D,E). Notably, the cytotoxicity of the DTYMK inhibitor turned out to be more toxic than pamiparib and other PARP1 inhibitors. The IC50 value was 7.49 µM for MP41 and 19.87 µM for MP46 cells (Figure 4F,G). Generally, in the XTT assay, the MP46 cell line was more resistant to the selected inhibitors (pamiparib and Ymu1) than the MP41 cells, and Ymu1 had a more prominent cytotoxic effect than pamiparib in both cell lines.

In the next step, we tested selected concentrations of pamiparib (P) and Ymu1 (Y) and their combinations on studied cells (Figure 5A,B). MP41 cells responded significantly to most of the tested concentrations of inhibitors, excluding only 5 µM pamiparib, 10 µM pamiparib, 5 µM Ymu1, and combinations of 5 µM pamiparib + 5 µM Ymu1 and 10 µM pamiparib + 5 µM Ymu1 (Figure 5A). While in MP46 cells, only four tested combinations had significant cytotoxic effects—5 µM pamiparib + 15 µM Ymu1, 10 µM pamiparib + 15 µM Ymu1, 25 µM pamiparib + 10 µM Ymu1, and the highest concentration of drugs—25 µM pamiparib + 15 µM Ymu1 (Figure 5B). The synergy evaluation [27] revealed distinctly different effects of the usage of the tested drug combinations on studied cell lines (Figure 5C,D). In the case of the cell line MP41, there is a synergy for the Ymu1 + pamiparib pair, as the Loewe synergy score was higher than 10. However, for MP46 cells, we noted the additive effect of the combination of those drugs because the Loewe score was around 2. In the end, we decided to use further on the following combinations of drugs: 10 µM pamiparib + 15 µM Ymu1, 25 µM pamiparib + 10 µM Ymu1, and 25 µM pamiparib + 15 µM Ymu1. Additionally, we also tested those drugs separately.

### 3.7. Effect of Pamiparib and Ymu1 and Their Combinations on EdU Incorporation 

To assess the proliferative capacity of MP41 and MP46 cells following treatment with selected inhibitors, we employed an EdU assay, which involves the incorporation of a thymidine analog (EdU) into the DNA of treated cells. The incorporation directly reflects the number of newly formed cells. For MP41 cells, the fluorescence intensity exhibited significant reduction compared to the vehicle control following the addition of 25 µM pamiparib, 15 µM Ymu1, and two tested combinations: 25 µM pamiparib + 10 µM Ymu1 and 25 µM pamiparib + 15 µM Ymu1 (Figure 6A). Particularly noteworthy is the observation that the amount of incorporated EdU to the DNA was the lowest in the sample with the highest combination of inhibitors (reduction by approx. 80% vs. vehicle control). Similarly, results for the MP46 cell line demonstrated a similar level of EdU in the cells upon adding drugs when compared to MP41 cells (Figure 6B). However, significant changes were observed in lower concentrations compared to the MP41 cell line; even a concentration of 10 µM pamiparib and the combination of 10 µM pamiparib with 15 µM Ymu1 exhibited a low amount of incorporated EdU. Interestingly, the combination of 10 µM pamiparib and 15 µM Ymu1 appeared to be more effective than the combination of 25 µM pamiparib and 10 µM Ymu1 in the MP46 cells. In both cell lines, there was a significant decrease in the amount of incorporated EdU when the cells were treated with 15 µM Ymu1, while 10 µM Ymu1 barely had any effect on the cells. Comparing the EdU incorporation assay with the XTT assay (Figure 5), we can see that, in some cases, there is a decrease in EdU incorporation, but this does not equal reduced cell viability. This is most likely due to fewer cells incorporating EdU into their DNA because the process of DNA synthesis is already inhibited by inhibitors or their combinations. However, this may not yet be mirrored.

### 3.8. The Influence of DTYMK and PARP1 Inhibitors on the mTOR Signaling Pathway

Usually, when investigating the proliferation rate and survival of cells in response to cytotoxic agents, the level of phosphorylated Akt1/2 (Protein Kinase B, PKB) is evaluated because it is a critical player in regulating cell proliferation [35]. Interestingly, the level of phosphorylated Akt1 is very low in the studied melanoma cell lines [32]. We studied UM cell lines characterized by high activity of the PI3K/mTOR signaling pathway [32]. Phosphorylation of S6 ribosomal protein (pS6) is downstream of mTOR activation and is a reliable surrogate marker of mTOR activity [36]. Thus, we decided to evaluate by Western blot technique the level of pS6 and S6 in MP41 and MP46 cells under tested conditions.

Lysates were collected at various time points, including 24, 48, and 72 h post-inhibitor treatment. Notably, lysates obtained after 72 h of inhibitor exposure exhibited considerable cell destruction, as indicated by a substantial accumulation of BSA protein on the resulting membranes. That is why we did not evaluate this time point. All analyzed proteins were detected in vehicle control samples, with their levels remaining consistent across these conditions and between cell lines (Figure 6C,D).

The total S6 protein level is relatively consistent and high in most conditions within the first 24 h of exposure to inhibitors, especially for MP46 cells, for which the effect of inhibitors on the S6 level was negligible. Interestingly, 24 h long exposure was sufficient for two combinations (25 µM pamiparib + 10 µM Ymu1 and 25 µM pamiparib + 15 µM Ymu1) to cause an exceptional reducing effect on the S6 level in MP41 cells (Figure 6C). For MP46 cells, those conditions were found similarly effective only after 48 h (Figure 6D). MP41 cells’ response to inhibitors after 48 h was analogous to those previously observed, but the trend included one more condition, which was found similarly effective to those already mentioned (10 µM pamiparib + 15 µM Ymu1). As for the phosphorylated form of S6 (pS6), there was a general trend of a prominent decrease in signal intensity observed after treatment of both cell lines with pamiparib and its combinations with Ymu1 for 24 and 48 h. It is noteworthy that Ymu1 alone tested in two concentrations did not affect the phosphorylation level of S6.

### 3.9. The Effect of Drug Treatment on Cells’ Motility Abilities and Their Morphology

We conducted a single-cell migration assay using the Incucyte^®^ system, capturing images of live cells at 2-hour intervals over 72 h. We routinely use this method to assess cells’ ability to migrate under 2-D conditions [37]. Analysis of captured images revealed interesting observations. Remarkably, just 24 h after administering the tested compounds, the number of cells lowered compared to non-treated cells for the MP41 cell line (Appendix A). The lowest cell count was observed following treatment with 15 µM Ymu1 and combinations: 10 µM pamiparib + 15 µM Ymu1 and 25 µM pamiparib + 15 µM Ymu1. However, in the case of MP46 cells, we did note changes in the number of cells after administration of the tested drugs and their combinations (Appendix A). Additionally, a noticeable change in cell morphology was observed as early as 1 h after adding the tested inhibitors, evident in both cell lines. Cells treated with 15 µM Ymu1 and combined with 10 or 25 µM pamiparib exhibited a rounded shape, and it stayed like that until the experiment’s end (Appendix A).

Based on the acquired images, we analyzed the distance covered by individual cells and their respective migration speeds (Figure 7). Moreover, we analyzed the trajectories of single-cell migration (Appendix A). Throughout 72 h, vehicle control MP41 cells covered an average distance of approx. 35 µm with an average speed of approx. 0.5 µm^2^/min (Figure 7A,B). In comparison, vehicle control MP46 cells covered an average distance of approx. 60 µm within the same timeframe, at an average speed of approx. 0.82 µm^2^/min (Figure 7D,E). We did not note any alterations in the distance and the speed of MP41 in response to the treatment with individually administered drugs or their combinations. Surprisingly, for MP41 cells, we noted that in some conditions, the cells did not move from the starting point as far as control cells (Appendix A), although they covered similar distances (Figure 7A,B). Thus, we decided to check the directionality of the cell movement. The MP41 cells were less persistent in their movement after treatment with 15 µm Ymu1 and a combination of 10 µM pamiparib + 15 µM Ymu1 compared to control cells (Figure 7C).

Notably, MP46 cells exhibited an overall greater migratory capacity than MP41 cells. Treatment of the cells with 15 µM Ymu1 and its combination with pamiparib (pamiparib 10 µM + Ymu1 15 µM, and pamiparib 25 µM + Ymu1 15 µM) led to shorter covered distances due to their diminished velocity (Figure 7D,E). However, only the administration of Pamiparib 10µM+ Ymu1 15 µM combination led to impaired directionality of MP46 cells (Figure 7F).

## 4. Discussion

While no studies have specifically combined DTYMK and PARP1 inhibitors, their individual functions imply a potential for synergistic or additive effects, as indicated in our analyses. The inhibition of DTYMK can potentially enhance cancer cell sensitivity to PARP1 inhibitors by disrupting pyrimidine metabolism and amplifying DNA damage in cancer cells [38]. However, combining these therapeutic agents in in vivo trials may be challenging. Toxicity is a primary concern. Combining PARP inhibitors with DNA-injury-inducing cytotoxic agents often increases normal tissue damage [39]. The same issue could potentially arise with the combination of DTYMK and PARP1 inhibitors. Therefore, careful dose optimization and scheduling would be necessary to minimize toxicity while maximizing the therapeutic effect. Another challenge is the development of drug resistance. Despite initial responsiveness, tumors often eventually acquire resistance to PARP inhibitors via various mechanisms. Such mechanisms include the restoration of the DNA repair process via HR or increased or decreased regulation of NHEJ repair. Other proteins that catch inhibitors before they reach their target, the PARP protein, may also be present in cells. Perhaps for one of the above reasons the PARP inhibitors used in our work had such a poor effect on the tested cell lines. The combination of DTYMK and PARP1 inhibitors may potentially overcome this resistance commonly observed with sole PARP1 inhibitor use [40], thereby increasing the effectiveness of the therapy. The additive and synergistic effects of these therapeutic agents, as demonstrated in our in vitro analysis, hold promise for future combined therapy, potentially enhancing efficacy and possibly reducing the replicative potential of tumor cells and their resistance to the treatment. Nowadays, there is active research and development in the treatment of metastatic uveal melanoma. Clinical trials are evaluating many targeted therapies and immunotherapies targeting UM. Drug combinations involving MEK (phase III) or PI3K/Akt inhibitors (phase Ib/II) are also being explored [41]. These findings indicate promising progress in our research as we focus on targeting the PI3K/Akt pathway.

Elevated DTYMK activity has been observed across several malignancies, encompassing breast cancer, adrenocortical carcinoma, renal clear cell carcinoma, bladder carcinoma, and thyroid cancer [42]. However, DTYMK has been most thoroughly investigated in the context of hepatocellular carcinoma (HCC) and lung adenocarcinoma, wherein researchers extensively assessed the impact of inhibiting DTYMK or its knockdown on cancer cell behavior [43,44]. In both instances, heightened DTYMK levels were correlated with deteriorated overall survival, disease progression, and increased proliferation of cancer cells [6,43]. Analogous conclusions can be drawn in the context of uveal melanoma (UM), where we found a positive correlation between higher DTYMK levels and elevated proliferation capacity of UM cells, as evidenced by increased mitotic index and heightened Ki-67 expression.

The examinations involving DTYMK knockdown or inhibition in the previously mentioned malignancies yielded particularly intriguing findings. It was revealed that inhibition effectively impedes tumor growth by reducing cell proliferation and renders them more susceptible to chemotherapy, especially with platinum analogs [6,7,44]. In the context of lung adenocarcinoma, knockdown additionally led to a decrease in the PI3K/AKT signaling pathway, which plays a pivotal role in regulating the cell cycle and is often overactive in cancer cells, thereby reducing apoptosis rates and promoting proliferation [44]. Furthermore, inhibition of the enzyme led to compromised migration capacity of lung cancer cells [6]. In our MP41 and MP46 cell lines, the inhibition of DTYMK and PARP1 demonstrated a similar association, resulting in reduced cell velocity and, consequently, shorter distances covered (Figure 7). Upon separately examining BAP1-positive and BAP1-negative cell populations, it was observed that BAP1-positive cells generally exhibit diminished migratory capabilities compared to their BAP1-negative counterparts, a finding corroborated by existing literature [45,46].

The importance of PARP1 appears to have been investigated more thoroughly than DTYMK in the context of UM. In a study conducted by Gajdzis et al. in 2021, it was suggested that PARP1 might serve as a promising target for tailored treatment in advanced UM, as heightened PARP1 activity was associated with a poorer overall survival [47]. We partially confirmed these findings, demonstrating the efficacy of PARP1 inhibitors in combination with DTYMK inhibitors in in vitro analysis (Figure 6). As previously indicated, the activity of PARP1 is particularly important in neoplasms where the homologous recombination (HR) DNA repair mechanism is compromised. Thus, its inhibition may find applications in ovarian, prostatic, or pancreatic carcinomas [10,12]. Evidence from the available literature and our study suggests the impairment of this mechanism in UM. Firstly, Doherty et al. demonstrated heightened non-homologous end-joining (NHEJ) alternative mechanisms in UM, suggesting impairment of the HR mechanism [48]. Secondly, we observed alterations in the quantity of incorporated EdU when testing the effects of pamiparib and Ymu1 inhibitors on cancer cell lines, with limited impact on cell numbers. This observation may result from disruptions in DNA metabolism and deficiencies in repair mechanisms, subsequently affecting EdU incorporation rates. Though the doubling times for both cell lines differed, with 110 h for the MP46 cell line, it was decided to treat the cells in various assays, usually for 72 h, as a compromise for the two studied cell lines. It is crucial to note here that we did not synchronize the cells, so while seeding the cells for the tests, the cells were in different phases of the cell cycle, meaning that even in the case of MP46 cells, at least half of the cells had to enter mitosis within 72 h of experiment duration. It is clear that both cell lines had impaired ability to synthesize DNA as the EdU incorporation assay showed significant decreases in DNA synthesis upon incubation of the cells with tested drugs for 72 h, meaning that a significant part of the cells had to go from G1 phase to S phase of the cell cycle.

As the inhibition of DTYMK and PARP1 was found to be effective in in vitro and in vivo trials in numerous malignancies due to their important function in maintaining genome stability, we suggest that the application of pamiparib together with Ymu1 may be beneficial for UM patients. We observed that simultaneously inhibiting these two proteins harmed cancer cells (BAP1-positive and BAP1-negative cell lines) concerning all tested parameters, which was not the case when the inhibitors were applied separately (Table 1). This was demonstrated by a decrease in EdU incorporation, indicating the reduced ability to multiply, and a decrease in S6 protein levels, which is linked to the mTOR signaling pathway and suggests decreased cell survival (Figure 6). Furthermore, cell motility was decreased when PARP1 and DTYMK were subjected to inhibition (Figure 7). In light of the obtained results, we suggest that there is a need for in vivo trials with the application of pamiparib and Ymu1 in UM patients, as this creates an opportunity for new tailored therapy that may improve OS and prognosis in this specific type of human melanoma.

In summary, overexpression of both DTYMK and PARP1 is noted in a subset of UM and correlated with the worst prognosis in our series of primary UM. Co-targeting of DTYMK and PARP1 affected all tested cytophysiological parameters of two UM cell lines in our study, raising the therapeutic potential of combined DTYMK and PARP1 inhibition (Table 1).

## 5. Conclusions

Combined targeting of DTYMK and PARP1 with Ymu1 and pamiparib, respectively, causes significant cytotoxic effects on uveal melanoma cancer cells regardless of BAP1 status. Our in vitro findings provide insights into novel therapeutic avenues for managing uveal melanoma, warranting further exploration in preclinical and clinical settings.

## Figures and Tables

**Figure 1 cells-13-01348-f001:**
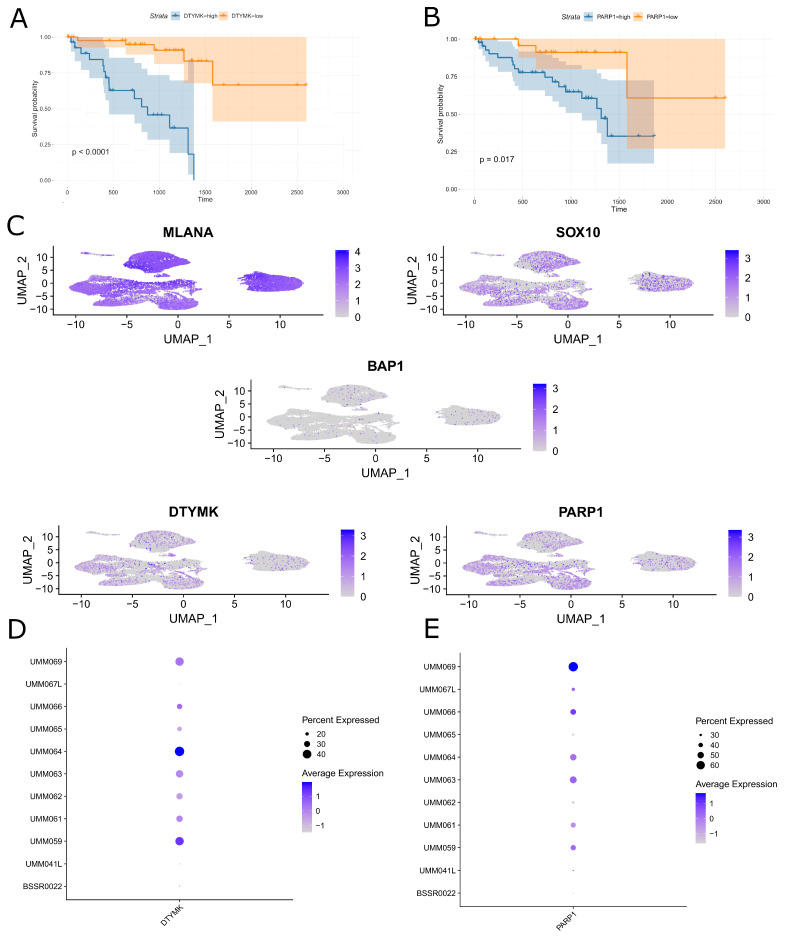
In silico analysis of *DTYMK* and *PARP1* RNA expression in UM. (**A**) Kaplan–Meier estimates of survival probability for UM patients on days from cancer diagnostics grouped by optimal cut point of *DTYMK* RNA expression level. (**B**) Kaplan–Meier estimates of survival probability for UM patients on days from cancer diagnostics grouped by optimal cut point of *PARP1* RNA expression level. (**C**) UMAP visualization of MLANA, SOX10, BAP1, DTYMK, and PARP1 expression in UVM tumor tissues (GSE139829). (**D**) Dot plot summarizing *DTYMK* expression in 11 UM samples included in GSE139829 dataset. Dot radius represents percent of tumors expressing given gene. Color intensity of dots reflects level of expression of given gene. (**E**) Dot plot summarizing *PARP1* expression in 11 UM samples included in GSE139829 dataset. Dot radius represents percent of tumors expressing given gene. Color intensity of dots reflects level of expression of given gene.

**Figure 2 cells-13-01348-f002:**
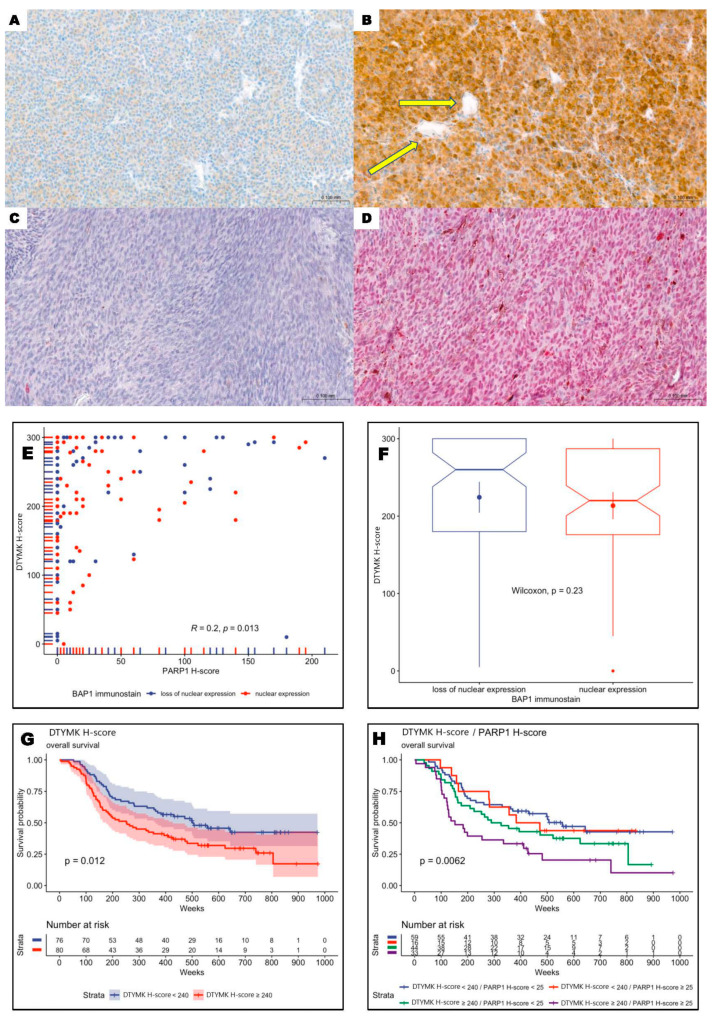
Immunohistochemistry of DTYMK and PARP1 in uveal melanoma patients and correlation between DTYMK and PARP1 H-scores. (**A**) Low cytoplasmic DTYMK expression in uveal melanoma cells (200×). (**B**) High DTYMK immunoreactivity in neoplastic cells. Tumoral vessels are negative (yellow arrows) (200×). (**C**) Lack of PARP1 expression in tumoral cells (200×). (**D**) Enhanced nuclear PARP1 reactivity in uveal melanoma cells (200×). (**E**) Plot showing DTYMK H-score values versus PARP1 H-score values (every point is a single patient). There is a very weak correlation (Pearson’s r = 0.2, *p*-value 0.013). (**F**) Dependency between DTYMK H-score and BAP1 box plots show the distribution of DTYMK H-score separated by BAP1 immunostaining. Wilcoxon rank-sum test (*p*-value = 0.23) shows no significant difference between values of DTYMK for patients with loss of nuclear expression and those with retained nuclear expression of BAP1 (**F**). (**G**) Kaplan–Meier analysis revealed a negative impact of enhanced expression of DTYMK as a single marker and (**H**) in combination with PARP1 on shorter overall survival of uveal melanoma patients. Patients with overexpression of both proteins were characterized by the worst long-term prognosis.

**Figure 3 cells-13-01348-f003:**
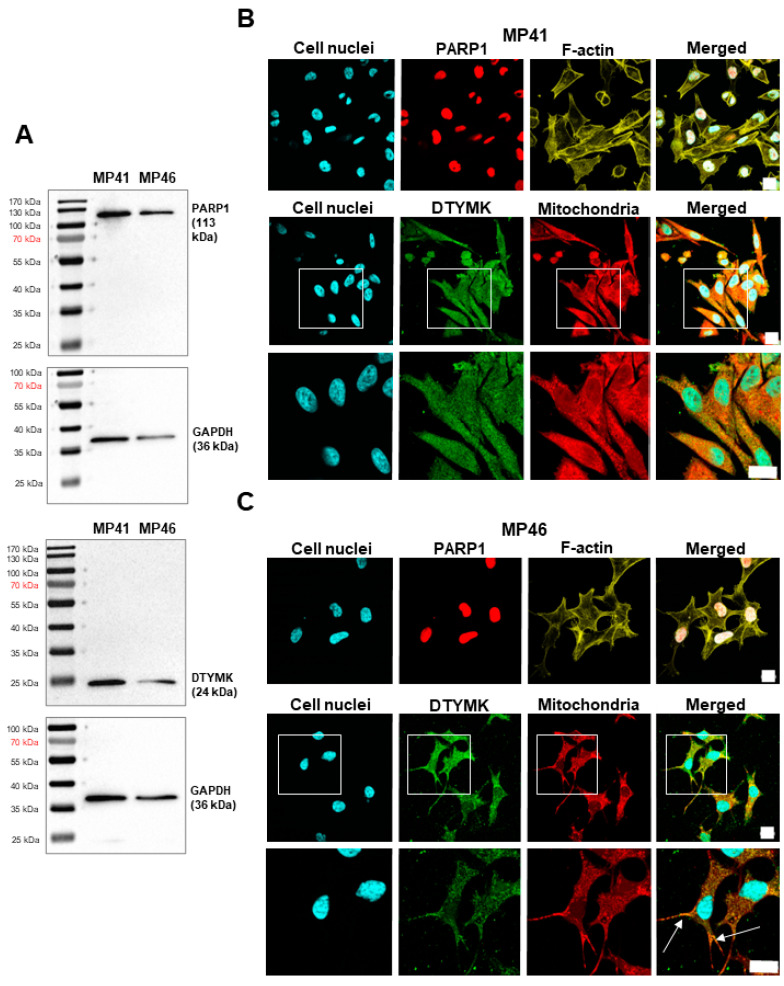
Detection of PARP1 and DTYMK and analysis of their localization in MP41 and MP46 cells. (**A**) Detection of PARP1 and DTYMK. A total of 15 µg of protein was loaded onto a single lane. Membranes were probed with specific antibodies to PARP1 and DTYMK. GAPDH was used as a sample loading control. (**B**,**C**) Analysis of PARP1 and DTYMK localization in MP41 (**B**) and MP46 (**C**) cells. Phalloidin-conjugated Alexa Fluor™ 568 labeled F-actin (yellow). Hoechst 33242 staining was used to visualize cells’ nuclei (blue), whereas MitoTracker™ labeled mitochondria (red). Cells were also incubated with antibodies directed to PARP1 and DTYMK and next with secondary antibodies conjugated with fluorophores: Alexa Fluor™ 647 (red) and Alexa Fluor™ 488 (green). The scale bar represents 20 µm. Arrows indicate particularly evident colocalization of DTYMK and mitochondria.

**Figure 4 cells-13-01348-f004:**
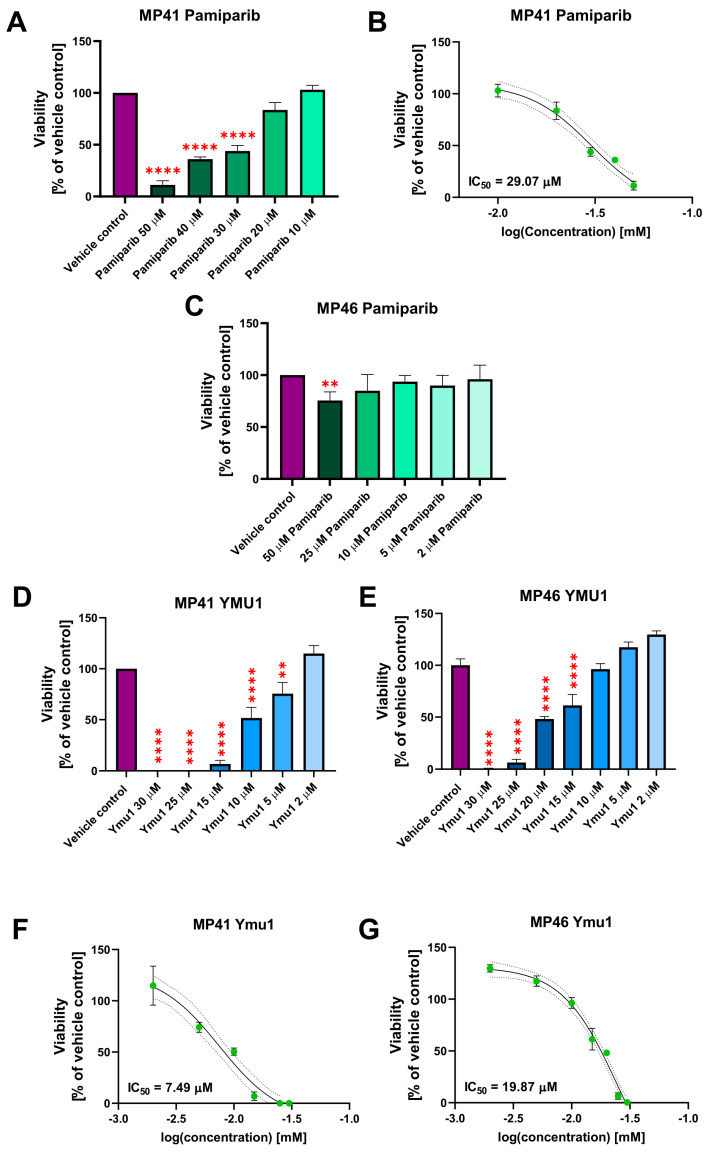
Cytotoxic effect of pamiparib and Ymu1 on MP41 and MP46 cell viability and its IC50 values. (**A**–**C**) Cytotoxic effect of pamiparib on MP41 (**A**) and MP46 (**B**) cells viability and its IC50 value in the range 10–50 µM (**C**). The same amount of DMSO was used for the vehicle control, which was present when cells were treated with 50 µM pamiparib. (**D**–**G**) Cytotoxic effect of Ymu1 on MP41 (**D**) and MP46 (**E**) cells viability and its IC50 values. (**F**,**G,** respectively) in the range 2–30 µM. The same amount of DMSO was used for the vehicle control, which was present when cells were treated with 30 µM Ymu1. Cells were incubated with inhibitors for 72 h and then subjected to the XTT assay. Results are presented as a mean (% of vehicle control) ± SD of four replicates. Asterisks above the bars express significance vs. vehicle control: *p* ≤ 0.01 (**) and *p* ≤ 0.0001 (****).

**Figure 5 cells-13-01348-f005:**
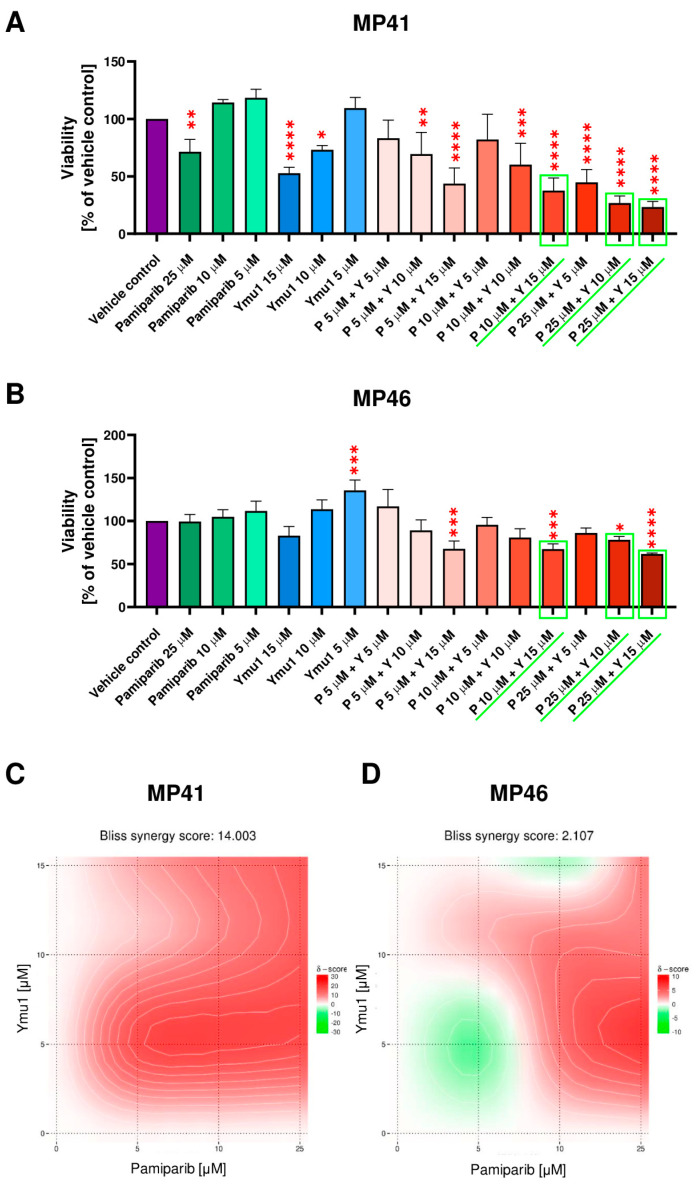
Cytotoxic effect of pamiparib and Ymu1 combinations on MP41 (**A**) and MP46 (**B**) cells viability and evaluation of synergy of examined drugs’ combinations for MP41 (**C**) and MP46 cells (**D**) calculated based on the results shown in (**A**,**B**). (**A**,**B**) For the vehicle control, the same amount of DMSO was used, which was present when cells were treated with 25 µM pamiparib (P) and 15 µM Ymu1 (Y). Cells were incubated with inhibitors for 72 h and then subjected to the XTT assay. The combinations of inhibitors selected for further studies are marked in green. Results are presented as a mean (% of vehicle control) ± SD of four replicates. Asterisks above the bars express significance vs. vehicle control: *p* ≤ 0.05 (*), *p* ≤ 0.01 (**), *p* ≤ 0.001 (***), and *p* ≤ 0.0001 (****). (**C**,**D**) Synergy scores’ interpretation: a score less than −10 indicates antagonistic interaction, a score between −10 and 10 implies additive interaction, and scores larger than 10 are typical for synergistic interactions. Analysis was performed using SynergyFinder software [27].

**Figure 6 cells-13-01348-f006:**
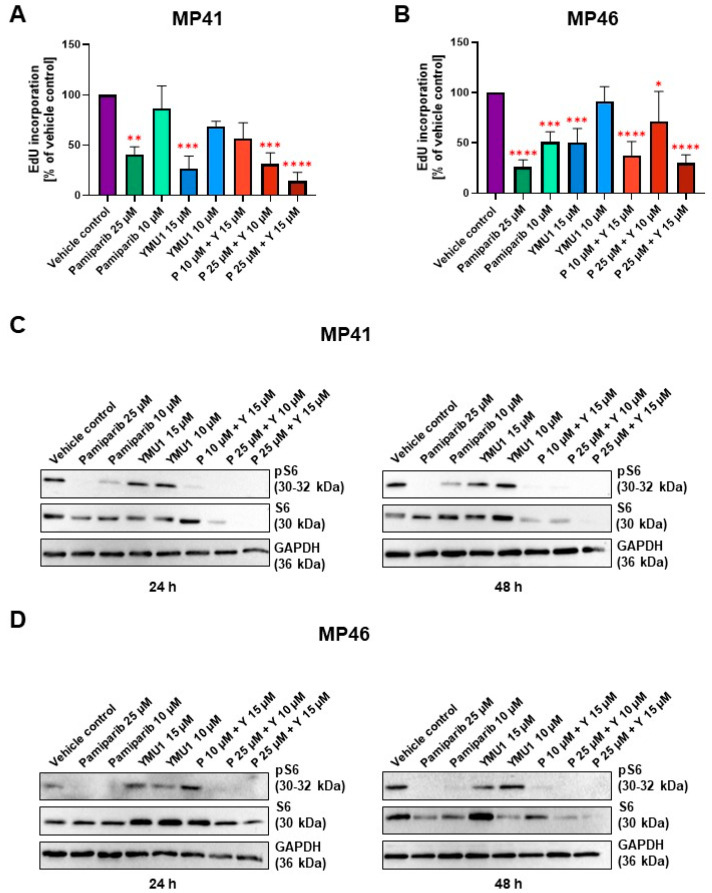
EdU incorporation for MP41 (**A**) and MP46 cells (**B**) after administration of selected tested inhibitor concentrations and the effect of pamiparib and Ymu1 on the amount and activity of S6 (**C**,**D**). (**A**,**B**) The same amount of DMSO was used for the vehicle control, which was present when cells were treated with 25 µM pamiparib (P) and 15 µM Ymu1 (Y). A total of 48 h after the addition of inhibitors to cells, EdU reagent was added to the cells for the next 24 h. After that, the cells were processed according to the manufacturer’s protocol to assess the amount of incorporated EdU. Results are presented as mean ± SD of three replicates. Asterisks above the bars express significance vs. vehicle control; *p* ≤ 0.05 (*), *p* ≤ 0.01 (**), *p* ≤ 0.001 (***) and *p* ≤ 0.0001 (****). (**C**,**D**) A total of 15 µg of proteins was loaded onto a single lane. Cells were incubated with indicated concentrations of inhibitors for 24 or 48 h. The same amount of DMSO was used for the vehicle control, which was present when cells were treated with 25 µM pamiparib (P) and 15 µM Ymu1 (Y). Membranes were probed with specific antibodies to total and phosphorylated forms of S6. Representative replicates are shown.

**Figure 7 cells-13-01348-f007:**
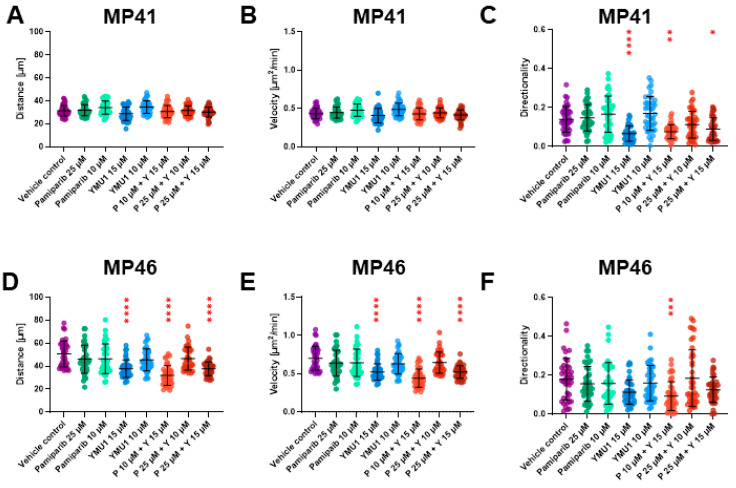
Distance, velocity, and directionality were calculated from spontaneous migration assay for MP41 (**A**–**C**) and MP46 (**D**–**F**) cells after treatment with selected inhibitors. The experiment was run for 72 h using the IncuCyte^®^ system. Images were collected every two hours, and after the experiment was ready, the trajectories of movement of single cells were prepared (40 cells per condition) (Appendix A). Based on the trajectories, distance, velocity, and directionality of motile cells were assessed (n = 3). The same amount of DMSO was used for the vehicle control, which was present when cells were treated with 25 µM pamiparib (P) and 15 µM Ymu1 (Y). Results are presented as mean ± SD. Asterisks above the bars express significance vs. vehicle control; *p* ≤ 0.05 (*), *p* ≤ 0.01 (**), *p* ≤ 0.001 (***) *p* ≤ 0.0001 (****).

**Table 1 cells-13-01348-t001:** A summary of results obtained from experiments on MP41 and MP46 cells. The red color indicates deteriorated parameters assessed in the listed assays. P—Pamiparib, Y—Ymu1.

	MP41 Cell Line (BAP-Positive, c.626al>A/T Mutation in the *GNA11* Gene)			MP46 Cell Line (BAP-Negative, c.626 A>T Mutation in the *GNAQ* Gene)		
	Pamiparib	YMU1	Combinations of Inhibitors	Pamiparib	YMU1	Combinations of Inhibitors
XTT assay	IC_50_ = 29.07 μM	IC_50_ = 7.49 μM	High cytotoxicity	No cytotoxicity	IC_50_ = 19.87 μM	Middle level cytotoxicity
Incoropration of EdU	Reduction by approx. 60% (P 25 μM)	Reduction by approx. 70% (Y 15 μM)	Reduction by approx. 60–80% (P 25 μM + Y 10 μM P 25 μM + Y 15 μM)	Reduction by approx. 50–70% (P 10 μM, P 25 μM)	Reduction by approx. 50% (Y 15 μM)	Reduction by approx. 70% (P 10 μM + Y 15 μM P 25 μM + Y 15 μM)
Western Blot analysis	Lack of phosphorylated S6	S6 is phosphorylated	Lack of phosphorylated S6	Lack of phosphorylated S6	S6 is phosphorylated	Lack of phosphorylated S6
Morphology	No changes	Cells are rounded (Y 10 μM, Y 15 μM)	Cells are rounded	No changes	Cells are rounded (Y 15 μM)	Cells are rounded (P 10 μM + Y 15 Μm P 25 μM + Y 15 μM)
Single-cell migration assay	No changes	Changed directionality (Y 15 μM)	Changed directionality (P 10 μM + Y 15 μM)	No changes	The covered distance is shortened, and the velocity is reduced (Y 15 μM)	The covered distance is shortened, and the velocity is reduced (P10 μM + Y 15 μM P 25 μM + Y 15 μM) Changed directionality (P 10 μM + Y 15 μM)

## Data Availability

The datasets used and/or analyzed during the current study are available from the corresponding author upon reasonable request.

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
