# Peer review of "Co-Targeting of DTYMK and PARP1 as a Potential Therapeutic Approach in Uveal Melanoma"

_cells, 2024, doi:10.3390/cells13161348_

Round 1
Reviewer 1 Report
Comments and Suggestions for Authors
In this manuscript, Ozieblo, Mizera et al. investigated co-inhibition of the two DNA metabolism enzymes PARP1 and DTYMK as a potential anti-tumor approach in uveal melanoma. Their research involved in silico analysis, clinical histology analyisis and in vitro experiments with two uveal melanoma cell lines, each exhibiting distinct BAP1 status. They observed that inhibitors pamiparib and Ymu1 reduced viability of cell lines to different degrees. Combined treatments showed synergistic or additive effects depending on cell line and condition. Furthermore, combined treatments reduced proliferation, cell mortility and migration speed. The hypothesis of „double hit into DNA metabolism“ in uveal melanoma was confirmed in vitro as a possible therapeutic option.
This comprehensive study suggests DNA metabolism enzymes involved in uveal melanoma pathology and thus as therapeutic targets. The novelty of the study lies in co-inhibition of PARP1 and DTYMK as most effective treatment option.
My comments and suggestions for the authors are:
1. The cell lines are well choosen. However, the data of the two cell lines were compared in a descriptive manner, as the results of each cell line were presented as one cell line per graph. In order to confirm the observed differences between the two cell lines MP41 and MP46, the data between the cell lines must be statistically analyzed.
2. Apart from the different genetic profile of the cell lines, there is another important difference between the cell lines to be discussed, namely the different doubling time of MP41 is 41h and that of MP46 is 110h. The authors choose 72h or 48h for drug testing. Within 72 or 48 hours, MP41 has already doubled while MP46 did not, which are completely different DNA replication conditions that could affect the effectiveness of the inhibition of DNA metabolic enzymes. The observed differences between the cell lines could simply be due to their different proliferation rate and thus different number of DNA replications during the course of the experiments?
3. In the introduction section, it remains unclear/mixed whether you have mentioned earlier work (please add a reference) or you have described present work in the manuscript (add e.g. “in the present study”):
Line 56-58: Based on our preliminary in silico analysis…targets (reference?/present study?)
Line 77-81: Our bioinformatical analysis…available (reference?/present study?)
Line 102-105: Our preliminary clinical studies…patients (reference Gajdzis et al, 2021?)
4. In the introduction section line 61, you mentioned the “hypothesis of the double hit into tumoral DNA metabolism”. Is that your own hypothesis or was it put forward earlier? Since the hypothesis appears central to your study, explain the hypothesis in more detail and/or add relevant references.
5. Methods –single cell migration assay: how many cells were seeded in the wells? Results – Figure S2: It appears that cell number were much lower in vehicle control MP46 compared to MP41, initial conditions appears different?
6. Results – Figure 1A, B and Figure 2 E,F,G,H: can be improved, the text in the figure is difficult to read. Figure legend 1: “expression levels” of RNA or protein expression?
7. Results – line 385: reference 34 is the same like reference 32 (references, line 743 versus 749)
8. Results - Figure 3A: PARP1 blot is only shown for MP41, DTYMK blot is only shown for MP46 – the blots are not labelled correctly?
9. Results –Figure S1: The study includes testing of several PARP1 inhibitors while only one DTYMK inhibitor Ymu1 was included. Did you also tested other DTYMK inhibitors on uveal melanoma cell lines and if so what was the outcome? Why are the other PARP1 inhibitors ineffective in lowering viability of MP41, if the inhibitors are also inhibiting the same enzyme? Why none of the PARP1 inhibitors showed cytotoxic effect on MP46 if this cell line also contains the same enzymes as the MP41 cell line?
10. Results- Figure 4C: 50µM pamiparib exerted no effect on viability of MP46, in contrast the same concentration of 50 µM statistically significant reduced viability in Figure S1, G. Are these effects reproducible?
11. Results- Figure 4D-G and Figure 5 B: Low Ymu1 concentrations of 2µM (Figure 4 D-G) or 5µM (statistically significant in Figure 5B) leads to higher viability than vehicle control, explain this unexpected findings.
12. Results – Figure 5C, D, Figure legend: Which samples/conditions have been included in the synergy analysis and are shown in the figure?
13. Results- Figure 6 B, MP46 cells: PARP1 inhibition with 25µM or 10µM pamiparib reduced EdU incorporation while there was no effect on viability (Figure 5B), comment on these findings.
14. Results- Figure 7: The figure could be improved, the red asterisks above the dots of bars are difficult to recognize. Figure 7 legend: How many cells have been analysed each bar?
15. Results –Figure 7 A, B, C versus Figure 7 D, E, F: Are the measured differences between the cell lines statistically significantly different?
16. Discussion –line 588-591: Analogous conclusions...Ki-67 expression. I can’t find the correlation between DTYMK and Ki-67 staining in the present study, add a Figure or a reference.
17. Discussion: If you mention your own results, refer to the figure in which the results are shown ("Figure xy") so that the reader can follow your discussion more easily.
18. Discussion –line 603-606: Upon separately…literature (ref 46). Reference 46 refers to MSCs, a reference with uveal melanoma cells or other cancer cells would be better.
19. Discussion-Table 1 can be improved; the text is difficult to read.
In this manuscript, Ozieblo, Mizera et al. investigated co-inhibition of the two DNA metabolism enzymes PARP1 and DTYMK as a potential anti-tumor approach in uveal melanoma. Their research involved in silico analysis, clinical histology analyisis and in vitro experiments with two uveal melanoma cell lines, each exhibiting distinct BAP1 status. They observed that inhibitors pamiparib and Ymu1 reduced viability of cell lines to different degrees. Combined treatments showed synergistic or additive effects depending on cell line and condition. Furthermore, combined treatments reduced proliferation, cell mortility and migration speed. The hypothesis of „double hit into DNA metabolism“ in uveal melanoma was confirmed in vitro as a possible therapeutic option.
This comprehensive study suggests DNA metabolism enzymes involved in uveal melanoma pathology and thus as therapeutic targets. The novelty of the study lies in co-inhibition of PARP1 and DTYMK as most effective treatment option.
My comments and suggestions for the authors are:
1. The cell lines are well choosen. However, the data of the two cell lines were compared in a descriptive manner, as the results of each cell line were presented as one cell line per graph. In order to confirm the observed differences between the two cell lines MP41 and MP46, the data between the cell lines must be statistically analyzed.
2. Apart from the different genetic profile of the cell lines, there is another important difference between the cell lines to be discussed, namely the different doubling time of MP41 is 41h and that of MP46 is 110h. The authors choose 72h or 48h for drug testing. Within 72 or 48 hours, MP41 has already doubled while MP46 did not, which are completely different DNA replication conditions that could affect the effectiveness of the inhibition of DNA metabolic enzymes. The observed differences between the cell lines could simply be due to their different proliferation rate and thus different number of DNA replications during the course of the experiments?
3. In the introduction section, it remains unclear/mixed whether you have mentioned earlier work (please add a reference) or you have described present work in the manuscript (add e.g. “in the present study”):
Line 56-58: Based on our preliminary in silico analysis…targets (reference?/present study?)
Line 77-81: Our bioinformatical analysis…available (reference?/present study?)
Line 102-105: Our preliminary clinical studies…patients (reference Gajdzis et al, 2021?)
4. In the introduction section line 61, you mentioned the “hypothesis of the double hit into tumoral DNA metabolism”. Is that your own hypothesis or was it put forward earlier? Since the hypothesis appears central to your study, explain the hypothesis in more detail and/or add relevant references.
5. Methods –single cell migration assay: how many cells were seeded in the wells? Results – Figure S2: It appears that cell number were much lower in vehicle control MP46 compared to MP41, initial conditions appears different?
6. Results – Figure 1A, B and Figure 2 E,F,G,H: can be improved, the text in the figure is difficult to read. Figure legend 1: “expression levels” of RNA or protein expression?
7. Results – line 385: reference 34 is the same like reference 32 (references, line 743 versus 749)
8. Results - Figure 3A: PARP1 blot is only shown for MP41, DTYMK blot is only shown for MP46 – the blots are not labelled correctly?
9. Results –Figure S1: The study includes testing of several PARP1 inhibitors while only one DTYMK inhibitor Ymu1 was included. Did you also tested other DTYMK inhibitors on uveal melanoma cell lines and if so what was the outcome? Why are the other PARP1 inhibitors ineffective in lowering viability of MP41, if the inhibitors are also inhibiting the same enzyme? Why none of the PARP1 inhibitors showed cytotoxic effect on MP46 if this cell line also contains the same enzymes as the MP41 cell line?
10. Results- Figure 4C: 50µM pamiparib exerted no effect on viability of MP46, in contrast the same concentration of 50 µM statistically significant reduced viability in Figure S1, G. Are these effects reproducible?
11. Results- Figure 4D-G and Figure 5 B: Low Ymu1 concentrations of 2µM (Figure 4 D-G) or 5µM (statistically significant in Figure 5B) leads to higher viability than vehicle control, explain this unexpected findings.
12. Results – Figure 5C, D, Figure legend: Which samples/conditions have been included in the synergy analysis and are shown in the figure?
13. Results- Figure 6 B, MP46 cells: PARP1 inhibition with 25µM or 10µM pamiparib reduced EdU incorporation while there was no effect on viability (Figure 5B), comment on these findings.
14. Results- Figure 7: The figure could be improved, the red asterisks above the dots of bars are difficult to recognize. Figure 7 legend: How many cells have been analysed each bar?
15. Results –Figure 7 A, B, C versus Figure 7 D, E, F: Are the measured differences between the cell lines statistically significantly different?
16. Discussion –line 588-591: Analogous conclusions...Ki-67 expression. I can’t find the correlation between DTYMK and Ki-67 staining in the present study, add a Figure or a reference.
17. Discussion: If you mention your own results, refer to the figure in which the results are shown ("Figure xy") so that the reader can follow your discussion more easily.
18. Discussion –line 603-606: Upon separately…literature (ref 46). Reference 46 refers to MSCs, a reference with uveal melanoma cells or other cancer cells would be better.
19. Discussion-Table 1 can be improved; the text is difficult to read.
Author Response
Dear Reviewer,
We appreciate your valuable comments. Please find our response attached as a docx file.
Sincerely
Piotr Donizy

Reviewer 2 Report
Comments and Suggestions for Authors
In the introduction, the authors must discuss the role of TDO in uveal melanoma. There are several reports like (10.3390/cancers12020405) which discusses the involvement of TDO in UM cases. TDO is a heme protein, heme insertion mechanism of TDO was recently discovered (10.1016/j.freeradbiomed.2022.01.008), (10.1016/j.jbc.2023.104753). Please include these references in your manuscript along with a short paragraph about TDO, tryptophan breakdown, kynurenine formation and immune suppression in UM cases.
Author Response
In the introduction, the authors must discuss the role of TDO in uveal melanoma. There are several reports like (10.3390/cancers12020405) which discusses the involvement of TDO in UM cases. TDO is a heme protein, heme insertion mechanism of TDO was recently discovered (10.1016/j.freeradbiomed.2022.01.008), (10.1016/j.jbc.2023.104753). Please include these references in your manuscript along with a short paragraph about TDO, tryptophan breakdown, kynurenine formation and immune suppression in UM cases
Response: Dear reviewer, we appreciate your valuable comment. We acknowledge the significance of TDO in the context of uveal melanoma. However, we believe that the inclusion of this topic is not pertinent to the specific focus of our article. Given the already complex nature of our content, introducing additional concepts such as TDO could potentially reduce the clarity and readability of the manuscript.
Round 2
Reviewer 1 Report
Comments and Suggestions for Authors
My suggestions were appropriately adressed.